# A Prospective Pilot Trial to Assess the Efficacy of Argatroban (Argatra^®^) in Critically Ill Patients with Heparin Resistance [note 1]

**DOI:** 10.3390/jcm9040963

**Published:** 2020-03-31

**Authors:** Mirjam Bachler, Tobias Hell, Johannes Bösch, Benedikt Treml, Bettina Schenk, Benjamin Treichl, Barbara Friesenecker, Ingo Lorenz, Daniel Stengg, Stefan Hruby, Bernd Wallner, Elgar Oswald, Mathias Ströhle, Christian Niederwanger, Christian Irsara, Dietmar Fries

**Affiliations:** 1Institute for Sports Medicine, Alpine Medicine and Health Tourism, UMIT - University for Health Sciences, Medical Informatics and Technology, 6060 Hall in Tirol, Austria; mirjam.bachler@tirol-kliniken.at; 2Department of Mathematics, Faculty of Mathematics, Computer Science and Physics, University of Innsbruck, 6020 Innsbruck, Austria; tobias.hell@uibk.ac.at; 3Department of General and Surgical Critical Care Medicine, Medical University of Innsbruck, 6020 Innsbruck, Austria; johannes.boesch@tirol-kliniken.at (J.B.); lepidopterasoul@gmail.com (B.S.); barbara.friesenecker@tirol-kliniken.at (B.F.); ingo.lorenz@tirol-kliniken.at (I.L.); dany_strike@hotmail.com (D.S.); stefan.hruby@kepleruniklinikum.at (S.H.); bernd.wallner@i-med.ac.at (B.W.); mathias.stroehle@tirol-kliniken.at (M.S.); dietmar.fries@tirol-kliniken.at (D.F.); 4Department of Anaesthesiology and Critical Care Medicine, Medical University of Innsbruck, 6020 Innsbruck, Austria; benjamin.treichl@tirol-kliniken.at (B.T.); elgar.oswald@tirol-kliniken.at (E.O.); 5Department of Pediatrics, Pediatrics I, Intensive Care Unit, Medical University of Innsbruck, 6020 Innsbruck, Austria; christian.niederwanger@tirol-kliniken.at; 6Central Institute for Medical and Chemical Laboratory Diagnostics, Medical University of Innsbruck, 6020 Innsbruck, Austria; christian.irsara@tirol-kliniken.at

**Keywords:** Argatroban, hemorrhage, critical illness, prophylactic anticoagulation, thrombosis, unfractionated heparin

## Abstract

The current study aims to evaluate whether prophylactic anticoagulation using argatroban or an increased dose of unfractionated heparin (UFH) is effective in achieving the targeted activated partial thromboplastin time (aPTT) of more than 45 s in critically ill heparin-resistant (HR) patients. Patients were randomized either to continue receiving an increased dose of UFH, or to be treated with argatroban. The endpoints were defined as achieving an aPTT target of more than 45 s at 7 h and 24 h. This clinical trial was registered on clinicaltrials.gov (NCT01734252) and on EudraCT (2012-000487-23). A total of 42 patients, 20 patients in the heparin and 22 in the argatroban group, were included. Of the patients with continued heparin treatment 55% achieved the target aPTT at 7 h, while only 40% of this group maintained the target aPTT after 24 h. Of the argatroban group 59% reached the target aPTT at 7 h, while at 24 h 86% of these patients maintained the targeted aPTT. Treatment success at 7 h did not differ between the groups (*p* = 0.1000), whereas at 24 h argatroban showed significantly greater efficacy (*p* = 0.0021) than did heparin. Argatroban also worked better in maintaining adequate anticoagulation in the further course of the study. There was no significant difference in the occurrence of bleeding or thromboembolic complications between the treatment groups. In the case of heparin-resistant critically ill patients, argatroban showed greater efficacy than did an increased dose of heparin in achieving adequate anticoagulation at 24 h and in maintaining the targeted aPTT goal throughout the treatment phase.

## 1. Introduction

Critically ill patients are highly susceptible to developing thrombosis as well as to bleeding. Preventive therapy must therefore effectively prevent thrombosis while minimizing the risk of bleeding. The use of unfractionated heparin (UFH) or low molecular weight heparin (LMWH) for the prevention of venous thromboembolism (VTE) is recommended in critically ill patients [1,2].

Although the current guideline recommends LMWH rather than UFH [2], the rationale of this recommendation is based on only two meta-analyses, that compare subcutaneously given UFH twice daily with LMWH [2,3,4]. However, intravenous administration of UFH has been shown to be superior to subcutaneous administration preventing venous thrombotic events (VTE) [5]. A retrospective study showed no difference between LMWH and intravenously given UFH in patients with pulmonary embolism (PE), with respect to bleeding and change of PE severity [6]. For the use in critically ill patients, continuous intravenous administration of UFH is very advantageous as it enables constant plasma level and efficacy, with the option of expeditious dose adjustments. Additionally, UFH can be completely reversed by the administration of protamine sulfate [7].

Nevertheless, unfractionated heparin (UFH) may become ineffective. This unresponsiveness to heparin despite high doses is called heparin resistance (HR) [7,8,9,10,11,12]. The incidence of HR depends on the underlying disease or may occur after surgical intervention. Recently, in patients undergoing coronary artery bypass graft (CABG) operations the incidence of HR has been shown to be about 3% [13]. Moreover, in patients with cardiopulmonary bypass (CPB) the reported incidence of HR can be up to 20–25% [14,15].

At any rate, when HR occurs, these patients receive insufficient anticoagulation, putting them at considerable danger. This holds especially true for analgo-sedated critically ill patients with a high thrombotic risk. In this specific patient population, the ineffectiveness of thromboembolic prophylaxis increases the risk for thromboembolic complications to about 40–80% [16,17]. This high risk emphasizes the need for effective antithrombotic prophylaxis.

Argatroban, a direct thrombin inhibitor, could serve as an alternative anticoagulant for continuous infusion. Only a few studies have investigated argatroban as an alternative strategy for patients with HR [18,19,20]. To our knowledge no clinical prospective trial has been performed to date to test whether the off-label use of argatroban in heparin-resistant critically ill patients in need of antithrombotic prophylaxis is effective and safe.

In this clinical trial, we sought to investigate whether anticoagulation using argatroban (Argatra^®^, Mitsubishi Tanabe Pharma GmbH, Düsseldorf, Germany) or an increased dose of UFH is effective in achieving the targeted activated partial thromboplastin time (aPTT) of more than 45 s in critically ill heparin-resistant patients within a reasonable time period.

## 2. Materials and Methods

### 2.1. Trial Oversight

The study protocol, including slight changes in the number of participating departments within the same hospital, starting dose of argatroban, secondary objectives, and reduction of visits in order to take patient blood management into account, was approved by the pertinent medically independent Ethics Committee of the Medical University of Innsbruck (UN4633_LEK/AN2013-0041 331), as well as by the national competent authority for Austria.

Since at the time of enrollment most of the patients were unable to give their consent, obtaining consent prior to study enrollment was waived by the pertinent Ethics Committee, when necessary. The written informed consent was then sought as soon as the patient regained his or her consent capacity or a legal representative was available. The clinical trial was conducted in accordance with the Declaration of Helsinki and Good Clinical Practice guidelines. This trial was registered on clinicaltrials.gov: NCT01734252 and on EudraCT: 2012-000487-23.

The sponsor of the clinical trial was the Medical University of Innsbruck (Austria). The study was funded by Mitsubishi Pharma Deutschland GmbH (Germany).

### 2.2. Trial Design and Participants

This clinical trial was a prospective, monocenter, randomized, open-labeled, two-arm parallel-group trial conducted in critically ill patients to compare the efficacy of prophylactic anticoagulation using argatroban compared to standard therapy in heparin-resistant patients. The anticoagulation was assessed by measuring aPTT and should be achieved within 7 h (±1 h) after administration of the study medication. Unfractionated heparin (UFH), which was administered in an increased dosage, was considered standard therapy.

Critically ill patients with and without sepsis [21] at risk for thrombosis or thromboembolic complications (aged 18–85 years) and in need of prophylactic anticoagulation (aPTT of more than 45 s) were included if persistent heparin resistance occurred. Heparin resistance was defined as the non-achievement of aPTT > 45 s with a heparin dosage of 1200 IU per hour during continuous infusion with a minimum duration of two hours [7], as confirmed by measuring aPTT during screening before inclusion.

The key exclusion criteria were the need for an aPTT of more than 60 s for any reason, active bleeding, or if the risk for bleeding exceeded the risk for thromboembolic event as anticipated by the physician. Moreover, inevitable lethal course, severe liver failure defined as a Quick value of the prothrombin time (PT) < 30%, pregnancy, or planned peridural or spinal anesthesia during the study led to exclusion.

The first patient was included on 29 July 2012 (first patient first visit; FPFV). Study duration was about three years and eight months. The trial ended on 06 April 2016 (last patient last visit; LPLV).

#### Randomization and Blinding

The randomization schedule used a 1:1 allocation ratio. The study-specific randomization code was generated by the Department of Medical Statistics, Informatics, and Health Economics of the Medical University of Innsbruck (Austria) according to a random permuted blocks method with varying block size. This study-specific randomization number was pre-printed on envelopes consecutively prepared for each patient.

The recruiting physician consecutively opened the pre-numbered envelope for each patient. The envelope contained the information on whether the patient was to be allocated to the group continuing with an increased dose of heparin (standard therapy) or to the group receiving argatroban.

### 2.3. Study Medication and Procedures

The standard therapy group continued to receive heparin at 1200 IU per hour, which was increased to a maximum heparin dose of 1500 IU per hour. The initial dosage of argatroban was determined by the treating physician and was adjusted according to aPTT. Treatment failure was defined as not reaching an aPTT of more than 45 s with a maximum dose of the respective medication. For this purpose, the maximum doses were 1500 IU per hour for heparin and 10 µg/kg/min for argatroban. The study drug was infused continuously until anticoagulation with UFH was no longer needed or the study medication had been administered for a maximum duration of seven days.

If the aPTT could not be achieved at 7 h (±1 h) in the UFH group with the maximum dose, the patients had to be switched to the argatroban group. The same was true when aPTT fell below the targeted aPTT at a later time during the study treatment phase despite a maximum dose of UFH. If the aPTT decreased below 45 s despite the maximum dose of argatroban, the patient had to drop out of the study. If aPTT was prolonged or any surgical or further intervention with an increased risk for bleeding was needed, administration could be interrupted or the study drug dose could be reduced until the aPTT was within the targeted time.

Study-specific blood samples were taken at baseline before study drug administration. Further samples were taken at 2 (Visit 2), 7 (Visit 3), and 24 (Visit 4) h as well as on day 3 (Visit 5) and day 5 (Visit 6) after initiation of the study drug. On day 7 of study treatment or if the study medication was terminated before day 7, blood samples were taken directly before discontinuation of the study drug (Visit 7) and for safety reasons at minimum 6 h (Visit 8) after discontinuation of the study medication. If a patient from the heparin group had to switch to argatroban treatment, the study procedures recommenced at baseline (under heparin treatment) and followed the visits according to the protocol. Additionally, after the study drug was discontinued, a radiologist performed a duplex ultrasound scan of the lower extremity veins to detect clinically apparent non-apparent venous thromboembolism.

### 2.4. Outcome Measures

The primary objective of this study was to investigate which medication showed significantly greater efficacy in terms of achieving prophylactic anticoagulation defined as an aPTT >45 s at 7 h (±1 h) of treatment. For measurement of aPTT the Pathromtin^®^ SL Reagens (Siemens Healthcare Diagnostics GmbH, Vienna, Austria) was used on a BCS XP (Siemens Healthcare Diagnostics GmbH, Vienna, Austria).

Secondary objectives were the achievement and maintenance of the target aPTT during the seven treatment days in the randomization groups. Also investigated was the effectiveness of argatroban in the patients with therapy failure in the heparin group who needed to switch to the argatroban treatment group.

The clinical safety endpoints were the bleeding events and thrombotic complications until the end of study drug administration plus triple half-life time for serious adverse events (SAEs) and serious adverse reactions (SARs). Additionally, bleeding and thrombotic complications were recorded after this time point, but not as SAE or SARs, until a duplex ultrasound examination could be conducted close to treatment completion.

### 2.5. Statistical Analysis

Due to the lack of prospective randomized clinical trials investigating the efficacy of argatroban in heparin-resistant critically ill patients, a small retrospective study was chosen for sample size calculation. Based on the results of a previous retrospective study of 15 patients suffering from multiple-organ dysfunction syndrome (MODS) with heparin resistance and the need for off-label use of argatroban [20], a sample size of 20 patients per group was calculated to assess the primary endpoint with 80% power to detect a difference of 40% in the percentage of patients achieving a prophylactic aPTT using Fisher’s exact test with 0.05 two-sided significance level. We assumed that 50% of the control group would achieve a prophylactic aPTT, and 90% of the argatroban group would achieve the target aPTT. To account for drop-outs (10%) in this patient population, namely 22 patients per group, a total of 44 patients were randomized. The small sample size means only a “pilot trial” was designed. No interim analysis was planned.

A mathematician not involved in the study procedures or patient assessment performed the statistical analyses of the results using R, version 3.4.2 (free software for statistical computing and graphics). All statistical assessments were two-sided and a significance level of 5% was used. The Wilcoxon rank sum test and Fisher’s exact test were applied to assess differences between intention-to-treat (ITT) groups. We present continuous data as medians (25th–75th percentile) and binary variables as no./total no. (%). We show effect size and precision with estimated median differences for continuous data and odds ratios (OR) for binary variables, with 95% Confidence Intervals (CI).

As determined by the study protocol, missing aPTT measurements after V2 were imputed using the last observation carried forward (LOCF) method. Stratified for ITT groups, the course of aPTT measurements from V1 (baseline) to V6 (day 5) is illustrated by the sequence of the median with corresponding 95% CIs. For aPTT measurements at V2 to V6, a linear mixed-effects model with random intercepts for patients, as well as time points and ITT group as fixed effects, was fitted and we provide the mean difference in aPTT as effect size with a 95% CI. In addition, a logistic mixed-effects model was fitted for the binary outcome of achieving aPTT for longer than 45 s; we provide an adjusted odds ratio as effect size, with a corresponding 95% CI. For patients switched to the heparin arm, we illustrate the course of aPTT measurements in the same manner, however for purely exploratory purposes.

## 3. Results

### 3.1. Participants

A total of 44 patients were enrolled in this clinical trial. According to the pre-computed randomization list, 22 patients were randomized to the heparin group (Group H) and 22 patients to the argatroban group (Group A). Two patients did not meet the inclusion criteria for heparin resistance, retrospectively resulting in a total study population of 42 critically ill patients as seen from the flow diagram (Figure 1). Group assignment gave well-balanced patient characteristics (Table 1).

The antithrombin levels with inhibition activity on thrombin and factor Xa are in median 72% and 78%, respectively. Of the patients 19.5% had antithrombin levels with inhibitory effect on thrombin below 60%, while only a small proportion (7.3%) of the patients had antithrombin levels with inhibitory effect on factor Xa below 60%, as seen in Table 2.

### 3.2. Efficacy Endpoints

In the argatroban group, 59% of the patients reached the primary endpoint of achieving the aPTT target of more than 45 s after 7 h as compared to the standard therapy group with a slightly lower rate (55%), as depicted in Figure 2. The proportion of patients reaching the primary endpoint did not differ significantly between the two treatment groups (*p* = 0.1000).

In the argatroban group 12/22 (55%) of the patients who achieved the primary endpoint reached the target aPTT already after 2 h, whereas only 7/20 (35%) of the successfully treated patients in the heparin group achieved an aPTT > 45 s; OR 2.19 (0.55 to 9.32), *p* = 0.2321.

At 24 h only 40% of the patients treated with heparin achieved or maintained an aPTT of more than 45 s, whereas 60% had fallen below the aPTT target at this time point. In the argatroban group, 86% achieved the targeted aPTT of more than 45 s. Only 14% of the patients in the argatroban group failed to achieve the targeted aPTT. However, these patients had quite low argatroban levels of 0.27 (0.14 to 0.29) µg/mL compared to 0.46 (0.32 to 0.66) µg/mL (*p* = 0.0443). At 24 h treatment with argatroban they showed greater efficacy than did heparin in critically ill patients (*p* = 0.0021).

Moreover, argatroban showed better maintenance of sufficient anticoagulation prophylaxis than did heparin. This is reflected by the median aPTT for the argatroban group, which is significantly longer than the heparin group from Visit 4 (24 h) to Visit 6 (day 5), and by the aPTT for the heparin group, which shortened from Visit 4 onwards and failed to reach the target aPTT, as seen in Figure 3.

For aPTT measurements at V2 to V6, a linear mixed-effects model was fitted with random intercepts for patients, as well as time points and ITT group as fixed effects. In the argatroban arm, aPTT measurements were 7.7 (2.5 to 12.9) seconds longer (*p* = 0.0096).

A respective logistic mixed-effects model was fitted for the binary outcome of reaching an aPTT longer than 45 s, resulting in an adjusted odds ratio of 13.6 (2.1 to 87.6), *p* = 0.0062.

In the heparin group 15 patients (75%) needed to switch to argatroban. Two of these patients did not receive argatroban and were lost to follow-up. Therefore, a total of 13 patients were switched. Of these 13 patients eight had to be switched to argatroban after Visit 3, three further patients after Visit 4, and another two patients after Visit 5, as seen in Figure 4. The two patients who were lost to follow-up also would have had to be switched after Visit 5.

Additionally, at Visit 4 one patient in the switch group had recovered and no longer needed further continuous anticoagulation and therefore prematurely finished the treatment period. At Visit 5 the same was true for two patients in the argatroban group, and at Visit 6 for two patients in the heparin and one patient in the switch group. At Visit 7 there were two cases each of premature treatment termination in the argatroban and in the switch group.

These patients show a pattern similar to that seen in the ITT analysis. From V4 (24 h) onwards, aPTT was significantly longer in the argatroban group (Figure 5).

As seen in Table 3, 17 serious adverse events (SAE) occurred, with five of the 17 counting as serious adverse reactions (SAR) since they may be associated with the study drugs. However, no significant difference was seen between the treatment groups in either the SAEs (*p* = 0.7539) or the SARs (*p* = 0.656).

Thrombotic complications occurred in five patients (11.9%); in the argatroban group 13.6% experienced thrombosis as compared to 10% of the heparin patients without statistical difference (*p* = 1). In 7.1% of the total patient population (*n* = 3) bleeding occurred without any significant difference. Bleeding in the heparin group occurred in 10% as compared to 4.6% in the argatroban group (*p* = 0.2207).

## 4. Discussion

Although argatroban was retrospectively seen to be an effective and safe alternative anticoagulant in patients with heparin resistance [20], this current clinical trial was the first study to evaluate argatroban as an alternative thromboprophylaxis strategy in patients with heparin resistance in a prospective randomized design.

In our patient population, antithrombin deficiency seemed to not be the main reason for heparin resistance in the majority of our patients. If antithrombin levels decreased to ≤60%, then the ineffectiveness of heparin was associated with antithrombin deficiency [22]. In our patient population though, median antithrombin levels with activity against thrombin and factor Xa were 72% and 78%, respectively. Nevertheless, chemical studies revealed that UFH increases the kinetics of antithrombin activity on thrombin more than on factor Xa [23]. Since the activity of antithrombin inhibition of thrombin was decreased in 19.5% of our patients, this antithrombin deficiency might contribute to the development of heparin resistance. Resolution of heparin resistance with administration of antithrombin is questionable, as antithrombin levels do not correlate well with heparin responsiveness [22,24,25] and in post-surgical critically ill patients the HR was not at all associated with low levels of antithrombin [20]. Also, Nicholson et al. found no difference in antithrombin levels between heparin-resistant and heparin-responsive patients [26].

With regard to the primary endpoint, argatroban did not show greater efficacy in achieving adequate anticoagulation after 7 h. However, a trend in reaching the targeted aPTT was seen in the argatroban group.

One reason for the modest argatroban result at 7 h might be attributed to its cautious use. Determining the appropriate starting dose, especially in critically ill patients, is the subject of ongoing discussion, resulting in a recommended lower starting dose and careful titration steps [27,28,29].

After 24 h, argatroban showed greater efficacy of the prophylactic anticoagulation treatment. Of the argatroban patients, 86% achieved the target aPTT compared to only 40% of those in the heparin group. At subsequent visits, the argatroban group was seen to be significantly more successful in achieving the targeted aPTT goal than was the heparin group. This is in accordance with a retrospective study showing that argatroban performed better in reaching a targeted aPTT over time than did heparin [18]. In our study, the average aPTT in the argatroban group was 7.71 (2.12 to 13.31) seconds longer than in the heparin group (*p* = 0.013).

A possible reason for the better performance of argatroban might be that UFH, contrary to argatroban, binds to many molecules, especially chemokines and cytokines [30], which are significantly increased in critically ill patients. This might be the reason for the heparin resistance in critically ill patients and the better performance of argatroban in this particular patient population.

Additionally, argatroban, unlike heparin, is able to inhibit thrombin that is already bound to fibrin [31]. This probably alleviates the feedback loop of thrombin in the plasmatic coagulation. The same mechanism enables argatroban to attenuate thrombin-induced platelet aggregation and amplification [32,33,34].

Even when a thrombus is formed under argatroban treatment, less factor XIII might be integrated into the clot, as activation is indirectly inhibited via argatroban [35]. This could result in a looser clot structure, with the clot being more permeable and therefore probably less resistant to fibrinolysis [36]. Argatroban is known to enhance fibrinolysis with regard to time to lysis and rate of lysis [35] and the influence on the clot structure might contribute to it.

Regarding the safety of the two therapies, the treatment groups showed no difference in the incidence rate of serious adverse events (SAE), and were mainly related to infections due to progression of the underlying disease as well as a worsening of respiratory function.

The overall bleeding rate in the argatroban group was similar to or lower than that found in other ICUs [27,37,38,39]. Our rate lies within the ranges reported for argatroban use in critical care with an occurrence from 0% to 25% [27,29,37,39,40]. However, our affected patients suffered mainly from traumatic spinal cord injuries (SCI), thus being especially prone to thromboembolic complications [16].

Clearly, more clinical trials are needed to demonstrate rare complications and further evaluate the safety profile of argatroban in special populations. For example, the use of argatroban in pregnant women might be necessary. Heparin resistance and heparin-induced thrombocytopenia (HIT) also affect pregnant women [41,42,43]. Argatroban appears to be effective and safe during pregnancy [41,43,44].

A possible limitation is the assessment of sufficiency of antithrombotic prophylaxis using aPTT, as this assay is very reagent specific. Secondly, the anti-IIa assay is the more appropriate tool for measuring the action of argatroban in the patient’s blood. However, this assay is not yet available in every hospital.

Furthermore, our small sample size has to be kept in mind and requires cautious interpretation of side-effects, especially bleeding and thrombotic complications on the basis of this study. However, the scarcity of literature encountered when designing the study led us to believe that this sample size would be suitable for such a first prospective clinical trial.

Although in patients with heparin-induced thrombocytopenia (HIT) argatroban was shown to be effective in terms of preventing thrombotic events and decreasing thrombosis-associated mortality [45,46,47,48,49] with no increased bleeding [18,37,50,51,52], both efficacy and safety have to be further evaluated in heparin-resistant patients without HIT. Nevertheless, our pilot trial not only shows that argatroban performs better in achieving and maintaining the target anticoagulation state, even with this small sample size, but it also provides data that are important for calculating proper sample size for larger clinical trials in this particular patient population.

## 5. Conclusions

Argatroban is a potent treatment alternative for thromboembolic prophylaxis in critically ill patients with heparin resistance. In summary, argatroban for thromboembolic prophylaxis was seen to be more efficacious than an increased dose of heparin in achieving adequate anticoagulation at 24 h and in maintaining the targeted aPTT goal throughout the treatment phase.

## Figures and Tables

**Figure 1 jcm-09-00963-f001:**
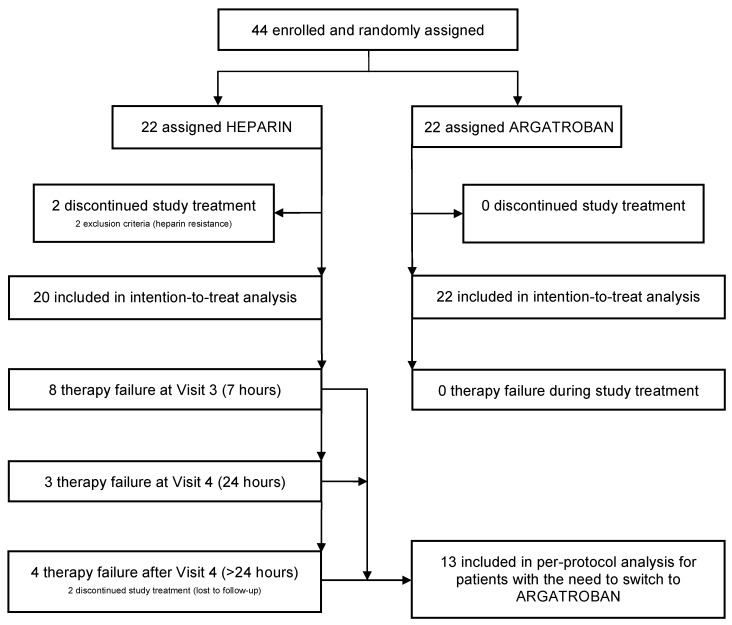
Trial profile Therapy failure in the heparin group = aPTT <45 s despite maximum heparin dose (1500 IU/h). Therapy failure in the argatroban group = aPTT <45 s despite maximum argatroban dose (10 μg/kg/min).

**Figure 2 jcm-09-00963-f002:**
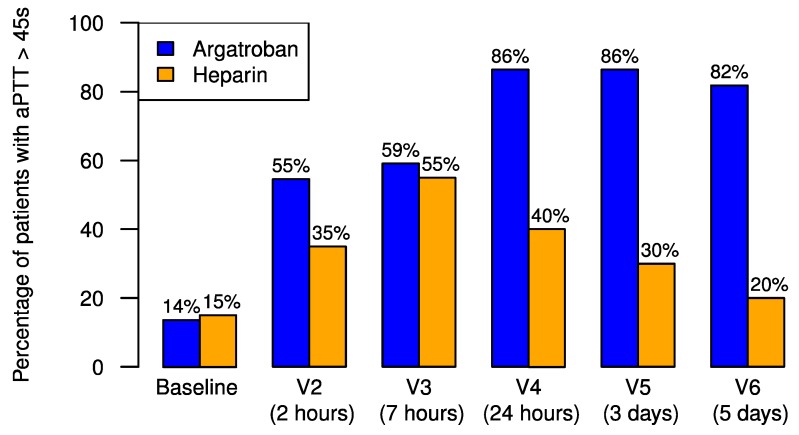
Achievement of the target activated partial thromboplastin time (aPTT) of more than 45 s. Patients who achieved the targeted aPTT of more than 45 s under heparin treatment (orange bars) and argatroban treatment (blue bars) in the intention-to-treat (ITT) groups. V indicates the study visits at pre-defined time points.

**Figure 3 jcm-09-00963-f003:**
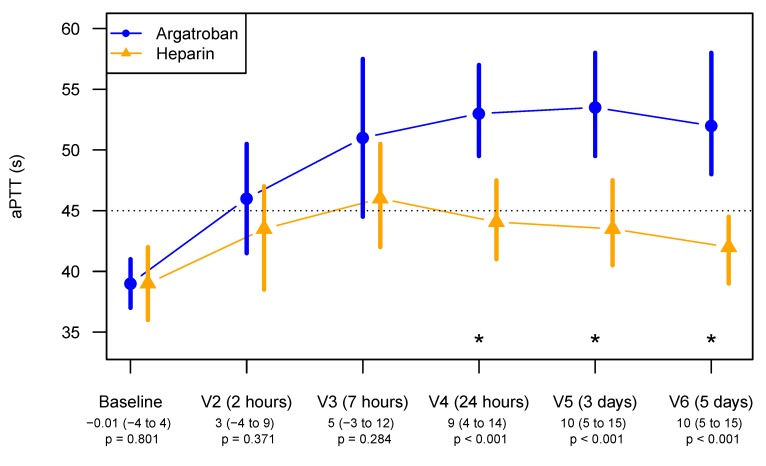
Median aPTT (95% CIs) during the study treatment from Baseline to Visit 6 (day 5) The dotted line at an aPTT of 45 s represents the targeted aPTT. Values below the visit labels indicate the estimated differences with 95% CIs between the aPTT for the argatroban and the heparin group. V indicates the study visits at pre-defined time points.

**Figure 4 jcm-09-00963-f004:**
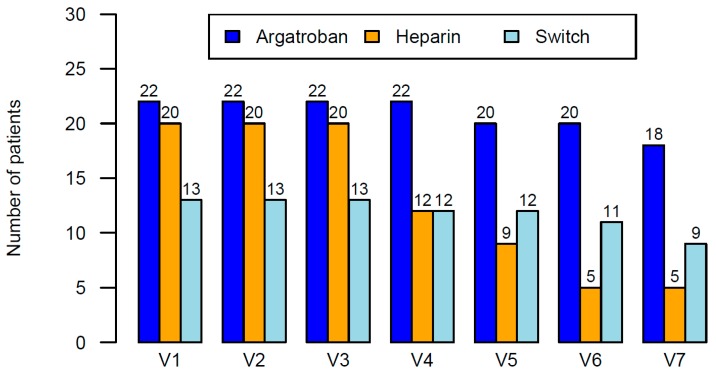
Number of patients per visit in the heparin, argatroban, and switch group. V indicates the study visits at pre-defined time points.

**Figure 5 jcm-09-00963-f005:**
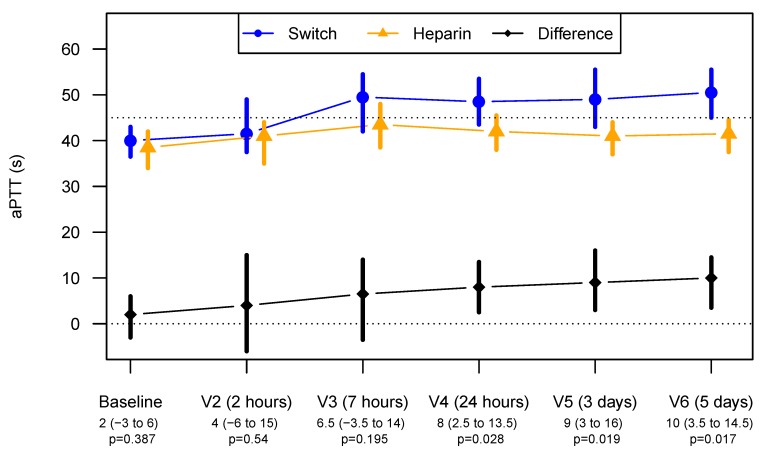
Median aPTT (95% CIs) during the study treatment from Baseline to Visit 6 (day 5) Depicted is the aPTT during treatment with heparin (orange) and argatroban (blue) after switch for only the patients who had to be switched. The upper dotted line at aPTT of 45 s represents the targeted aPTT. The black line indicates the difference between the aPTT during administration of heparin or argatroban in the patients that had to switch. Values below the visit labels indicate the estimated differences with 95% CIs between the aPTTs. V indicates the study visits at pre-defined time points.

**Table 1 jcm-09-00963-t001:** Characteristics ^a^ of patients.

	Total (*n* = 42)	A (*n* = 22)	H (*n* = 20)	Estimate ^b^ with 95% CI	*p* Value ^c^	Not Known ^d^
Weight (kg)	87 (81–105)	85 (80–93)	91 (85–105.75)	–6 (–17 to 3)	0.1846	1/2
Gender (female/male)	3/39	1/21	2/18	0.44 (0.01 to 9.05)	0.5976	0/0
Age (years)	56.5 (45–66.8)	56.5 (44.2–64.8)	57.5 (46–68)	–3 (–13 to 9)	0.5622	0/0
**Reason for hospitalization**						
(Poly)trauma	11/42 (26.2%)	6/22 (27.3%)	5/20 (25%)	1.12 (0.23 to 5.73)	1	0/0
Cardiovascular disease	15/42 (35.7%)	7/22 (31.8%)	8/20 (40%)	0.71 (0.16 to 2.97)	0.7488	0/0
Thromboembolic event	7/42 (16.7%)	5/22 (22.7%)	2/20 (10%)	2.59 (0.36 to 30.66)	0.4143	0/0
Oncologic patient with surgical intervention	3/42 (7.1%)	0/22 (0%)	3/20 (15%)	0 (0 to 2.12)	0.0993	0/0
Infectious disease	3/42 (7.1%)	2/22 (9.1%)	1/20 (5%)	1.87 (0.09 to 117.81)	1	0/0
Other	3/42 (7.1%)	2/22 (9.1%)	1/20 (5%)	1.87 (0.09 to 117.81)	1	0/0
**Reason for admission to intensive care unit**						
Postoperative: Cardiac surgery	4/42 (9.5%)	3/22 (13.6%)	1/20 (5%)	2.93 (0.21 to 165.47)	0.6079	0/0
Postoperative: Aortic surgery	8/42 (19%)	4/22 (18.2%)	4/20 (20%)	0.89 (0.14 to 5.64)	1	0/0
Postoperative: Gastrointestinal surgery	6/42 (14.3%)	4/22 (18.2%)	2/20 (10%)	1.97 (0.25 to 24.37)	0.6653	0/0
Polytrauma	9/42 (21.4%)	4/22 (18.2%)	5/20 (25%)	0.67 (0.11 to 3.77)	0.7139	0/0
Cardiac failure	3/42 (7.1%)	1/22 (4.5%)	2/20 (10%)	0.44 (0.01 to 9.05)	0.5976	0/0
Respiratory failure	5/42 (11.9%)	3/22 (13.6%)	2/20 (10%)	1.41 (0.14 to 18.72)	1	0/0
Renal failure	3/42 (7.1%)	2/22 (9.1%)	1/20 (5%)	1.87 (0.09 to 117.81)	1	0/0
Thrombectomy	2/42 (4.8%)	1/22 (4.5%)	1/20 (5%)	0.91 (0.01 to 74.67)	1	0/0
Other	2/42 (4.8%)	0/22 (0%)	2/20 (10%)	0 (0 to 4.79)	0.2207	0/0
History of thromboembolic events	15/42 (35.7%)	8/22 (36.4%)	7/20 (35%)	0.94 (0.22 to 3.98)	1	0/0

^a^ Binary data are presented as no./total no. (%), continuous data as medians (25th to 75th percentile), ^b^ Odds ratios for binary variables and estimated median difference (A–H) for continuous variables, ^c^ Assessed with Fisher’s exact test for categorical variables and the Wilcoxon rank sum test for continuous variables, ^d^ Not assessable in the argatroban/heparin group.

**Table 2 jcm-09-00963-t002:** Disease severity and baseline laboratory values ^a.^

	Total (*n* = 42)	A (*n* = 22)	H (*n* = 20)	Estimate ^b^ with 95% CI	*p* Value ^c^	Not Known ^d^
SIRS ^e^	41/42 (97.6%)	21/22 (95.5%)	20/20 (100%)	Inf ^f^ (0.02 to Inf)	1	0/0
Sepsis [21]	33/42 (78.6%)	17/22 (77.3%)	16/20 (80%)	1.17 (0.21 to 7.04)	1	0/0
SOFA ^g^ (pts)	7 (5–9)	7 (5–9.75)	7 (6–8.5)	–1 (–2 to 1)	0.5621	0/1
SAPS ^h^ II (pts)	38.5 (28.25–47.5)	38.5 (26.25–47.5)	37.5 (29.75–45.5)	–2 (–11 to 8)	0.6684	0/0
SAPS ^h^ II predicted mortality (%)	14.55 (7.15–28.05)	14.95 (6.45–28.02)	14.55 (7.25–32.25)	–0.3 (–12.7 to 8.9)	0.9198	0/0
SAPS ^h^ 3 (pts)	54.5 (47.25–62)	57.5 (46.25–64.25)	54 (47.75–56.75)	3 (–6 to 10)	0.5793	0/0
SAPS ^h^ 3 predicted mortality (%)	25 (13.5–40)	31 (12.5–44.5)	24 (14.5–29.5)	4 (–8 to 17)	0.5793	0/0
aPTT ^i^ (sec)	39 (36.25–43)	39 (37–41)	40.5 (35.5–43)	0 (–4 to 5)	1	0/0
Antithrombin FIIa	72 (64–81)	71 (57.25–78.75)	72 (64.5–83.5)	–6 (–13 to 6)	0.32	0/1
Antithrombin FXa	78 (69–89)	72 (67–86)	84.5 (75.75–98.25)	–10 (–20 to 0)	0.0503	1/0
Antithrombin FIIa <60	8/41 (19.5%)	7/22 (31.8%)	1/19 (5.3%)	0.12 (0 to 1.15)	0.0497	0/1
Antithrombin FXa <60	3/41 (7.3%)	2/21 (9.5%)	1/20 (5%)	0.51 (0.01 to 10.54)	1	1/0
C-reactive protein (mg/dL)	10.29 (6.77–19.74)	10.07 (6.9–19.8)	12.9 (5.69–17.11)	0.73 (–5.06 to 5.35)	0.8462	0/1
Procalcitonin (µg/L)	0.46 (0.25–1.24)	0.46 (0.26–1.52)	0.46 (0.26–1.17)	0 (–0.34 to 0.35)	1	0/1
FVIIa (%)	91 (75–113)	77 (69–119)	96.5 (81.5–110.75)	–11 (–26 to 9)	0.2507	1/0
FVIIIa (%)	279 (229–301)	282 (234–319)	275 (225–296.25)	21.05 (–15 to 60)	0.1966	1/0
vWF	347 (281.5–454.5)	342 (282–454)	349 (302–422)	–3 (–74 to 88)	0.9137	1/1
Bilirubin (mg/dL)	0.85 (0.53–1.53)	0.68 (0.4–1.63)	0.87 (0.75–1.34)	–0.2 (–0.52 to 0.14)	0.1956	0/1

^a^ Binary data are presented as no./total no. (%), continuous data as medians (25th to 75th percentile), ^b^ Odds ratios for binary variables and estimated median difference (A–H) for continuous variables, ^c^ Assessed with Fisher’s exact test for categorical variables and the Wilcoxon rank sum test for continuous variables, ^d^ Not assessable in the argatroban/heparin group, ^e^ Systemic Inflammatory Response Syndrome (SIRS), ^f^ Infinite (Inf), ^g^ Sequential Organ Failure Assessment (SOFA), ^h^ Simplified Acute Physiology Score (SAPS), ^i^ Activated partial thromboplastin time (aPTT).

**Table 3 jcm-09-00963-t003:** Outcome ^a^ of patients.

	Total (*n* = 42)	A (*n* = 22)	H (*n* = 20)	Estimate ^b^ with 95% CI	*p* Value ^c^	Not Known ^d^
Serious adverse events (SAE)	17/42 (40.5%)	8/22 (36.4%)	9/20 (45%)	0.7 (0.17 to 2.85)	0.7539	0/0
Serious adverse reactions (SAR)	5/42 (11.9%)	2/22 (9.1%)	3/20 (15%)	0.57 (0.04 to 5.64)	0.656	0/0
Thromboembolic event	5/42 (11.9%)	3/22 (13.6%)	2/20 (10%)	0.71 (0.05 to 6.97)	1	0/0
Bleeding event	3/42 (7.1%)	1/22 (4.6%)	2/20 (10%)	0 (0 to 42.9)	1	0/0
Ventilation	41/42 (97.6%)	21/22 (95.5%)	20/20 (100%)	Inf (0.02 to Inf)	1	0/0
Ventilator days	10 (6–18)	9 (6–16)	10.5 (6–18.25)	–1 (–6 to 4)	0.7241	1/0
Extracorporeal membrane oxygenation (ECMO)	4/42 (9.5%)	2/22 (9.1%)	2/20 (10%)	1.11 (0.07 to 16.77)	1	0/0
Renal replacement therapy (RRT)	15/41 (36.6%)	9/21 (42.9%)	6/20 (30%)	0.58 (0.13 to 2.47)	0.5204	1/0

^a^ Binary data are presented as no./total no. (%), continuous data as medians (25th to 75th percentile), ^b^ Odds ratios for binary variables and estimated median difference (A–H) for continuous variables, ^c^ Assessed with Fisher’s exact test for categorical variables and the Wilcoxon rank sum test for continuous variables, ^d^ Not assessable in the argatroban/heparin group.

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
