# Peer review of "A Prospective Pilot Trial to Assess the Efficacy of Argatroban (Argatra®) in Critically Ill Patients with Heparin Resistance"

_jcm, 2020, doi:10.3390/jcm9040963_

Round 1

Reviewer 1 Report

In this pilot study Bacheloret al  try to assess the efficacy of argatroban in critically ill patients with heparin resistance

This is a very interesting study, well written, with excellent experimental design. The results are very interesting however there are some points to be clarified:
1) The main concern is on the choice of sample size. The choice refers to a sample analyzed in a previous study that is not prospective randomized but retrospective. This is a potential bias.In other words, the number of patients chosen seems to be small to see a real difference in the two arms.The authors should specify these limits of the study in a more incisive way both in the methodological part and in the discussion.
2) It would also be interesting to evaluate an eventual AT III deficiency in patients with heparin resistance
3) Another problem is safety. To have a real therapeutic option it is necessary to better  evaluate the side effects (especially bleeding and thrombotic complications) ) in a larger sample of patients.
4) The authors should try to better explain the mechanism insight in the discussion for greater efficacy of argatroban vs UHF

Author Response

Answer 1: The reviewer adds an important point and is completely right. At the time of study design there was no prospective study done in heparin resistant critically ill patients. Therefore, we have chosen to only conduct a “pilot trial” in order to get sufficient data for sample size calculation for further clinical trials in a larger scale if needed.

We have added your valuable input in our manuscript accordingly in the method section (see page 4, line 169ff) and in the discussion section (see page 11, line 361ff).

Answer 2: The reviewer highlights an important point. The participation of antithrombin deficiency in the heparin resistance of our patients is indeed interesting and we have now added this analysis in the manuscript. For making the table 1 more readable, we have also divided table 1 into two separate tables with baseline characteristics and laboratory parameters. The information on antithrombin is now shown in table 2 (see page 6, line 220ff).

Although the median antithrombin levels with activity on thrombin and FXa are both above 70%. Nevertheless, an antithrombin deficiency of AT(FIIa) might play a role in our patients, since 19.5% experiencing such a deficiency at baseline. It is important to know, that UFH increases AT activity against IIa more than the activity against Xa.

We have now discussed the influence of antithrombin deficiency on heparin resistance in detail in the discussion section of the manuscript (see page 10, line 307ff).

Answer 3: The reviewer rightly criticizes the sample size. Here we found argatroban to be a safe alternative in regard to bleeding or thrombotic events though our small sample size is clearly underpowered for our results regarding safety.

Clearly, more and larger clinical trials are needed to further investigate other potential and rare side effects. We have emphasized the reader to interpret the safety data with caution due to the sample size in the discussion section. We have further highlighted the importance of larger clinical trials to prove the safety of argatroban also in this particular patient population, namely heparin resistant patients without HIT (see page 11, line 353ff and line 361ff).

Answer 4: The reviewer correctly points out a vague description of the mechanism. In the discussion section we have added the detailed action of argatroban, which add to the greater efficacy of argatroban and we also pointed out, that especially in critically ill patients, chemokines and cytokines are increased and UFH, in contrary to argatroban, tend to bind to these molecules (see page 11, line 332ff).

Reviewer 2 Report

There is an original and interesting clinical study that can be published after adding some more infomations with the appropriate references regarding the safety of Argatroban in pregnancy.

Author Response

Answer: We would like to thank this reviewer for the suggestion of adding information about safety in pregnancy. Indeed, it is true that pregnant women may develop HIT or heparin resistance and are therefore in need of alternative anticoagulation such as argatroban. We have added systematics retrospective analysis/reviews (unfortunately clinical trials are missing in this particular patient population) in the manuscript to point out, that argatroban is also a safe and effective alternative in pregnant women (see page 11, line 353ff).

Reviewer 3 Report

  1. trial design mentions that the efficacy of prophylactic anticoagulation was assessed; however, looking at patient characteristics in table 1 (by the way: top horizontal bar does not correspond to the columns underneath) I get the impression that part of patients were probably treated for thromboembolism, while others received anticoagulation indeed prophylactically (infectious diseases, oncologic patients with surgery...?). Can the authors clarify what was done exactly? 
  2. Looking at table 1 I think it should be reorganised as under the heading "Reason for admission to intensive care unit" a mixture of data is given including indeed reason for admission, but also classification of severity and a number of biomarkers; this is confusing. Divide in separate tables? 
  3. according to the definition used heparin resistance was diagnosed when a prolongation of aPTT > 45 secs was not achieved within 2 hours; however, fig. 2 shows that at 2 hours (after randomisation) 35% of patients on UFH had an aPTT > 45 secs? I may be confused and maybe baseline in this figure represents a 2 hour time slot instead? Please clarify. 
  4. Since 75% of patients on UFH switched to argatroban, the proportion of patients that responded well and completed the observation study is quite small. Still, it seems outcomes are recorded based on the original drug assignment (ITT), but how fair is that if the majority of patients switched drug along the line? Either way, the numbers in table 2 do not make sense to me, as for the A patients the outcomes are divided over the total population (42), whereas for UFH the number is divided by 22? How to explain these numbers? Also, I cannot trace the total numbers of SAR and SAE from the single outcomes? How does it all add up? 
  5. Fig. 3 probably  includes all subjects that at that moment were assigned to the respective drug: since there were 75% switchers from UFH to A, the numbers of patients must change over time, could this be indicated? Since the remainder of patients on UFH must have been "responders" I fail to understand why in general the aPTT values decline over time? Was this due to maximum doses UFH reached?
  6. Given all uncertainties I find the use of the word "superiority" for A over UFH not entirely appropriate.Eg in rule 275 the suggestion is made that A was superior in efficacy; this is clearly not a conclusion that can be drawn based on this small study. In terms of safety, the absolute number of bleedings seems higher in the A group, but again, this table 2 is confusing, as indicated. 
  7. Rules 294,295 seem to contain a fragment of a previous reviewer's comment? 

Author Response

Answer 1: The reviewer rightly adds an important point. 7 out of 42 subjects were hospitalized due to thromboembolic events. These thrombi were surgically /interventional removed and treatment of thromboses was completed before study inclusion. Our ICUs only treat postsurgical and trauma patients.

According to an institutional protocol on the two contributing ICUs UFH has been used for prophylactic anticoagulation (ie. an aPPT > 45 sec). Moreover, we excluded all patients in need for treatment of thromboembolism (ie. the need an aPTT > 60 sec).

Thank you for the hint, that the top horizontal bar is not fitting. We have formatted the tables.

Answer 2: We thank the reviewer for this excellent suggestion. We have now divided the first table into table 1 and 2. Former table 2 has been re-numbered into table 3.

Answer 3: You are completely right as this may be confusing. The heparin resistance (not achievement of aPTT > 45 sec within 2 hours) had to be diagnosed before study inclusion.

Before baseline we have done an aPTT for screening purposes and when aPTT was below < 45 seconds (and 1,200 IU heparin was given for a minimum of 2 hours) the patient was enrolled in the study. We have clarified this in the method section, that we used a screening aPTT for confirmation of the heparin resistance (see page 3, line 128).

But in few cases the aPTT increased and resulted in >45 seconds without any change of the study drug dose at the baseline measurement. From our clinical experience aPTT measurements may vary when repeatedly measured. These patients normally do not recover from heparin resistance without any change and may fall back below 45 seconds very soon. Therefore, these patients remained in the study and were kept diagnosed with heparin resistance. Anyhow, the percentage of these cases was the same in both randomization groups.

Two hours after the start of study drug aPTT was measured (Visit 2) for adjustment reasons. The study drug was either Argatroban or an increased dose of heparin (up to a maximum of 1.500 IE/h). Therefore, in figure 2 the percentage of patients reaching aPTT levels above 45 seconds increased in both groups as compared to baseline at two hours after start of study medication.

Answer 4: The reviewer correctly adds the point of an unprecise table 2 (now table 3). We have reformatted the corresponding table (see page 10, line 321ff).

You are right that you cannot trace the SARs from the single outcomes thromboembolic events and bleeding. This is explained by the fact, that we have set the reporting period for SARs such as bleeding and thrombotic events until the end of study treatment plus 3-times of the maximum half-life time of the study drugs. The end of this reporting period was reached with the conduction of visit 8 (not shown in the manuscript because it was only done for safety reasons).

But few events were happening after the conduction of visit 8 (conduction of an safety duplex sonography) and we didn’t want to hide these events from the readers. We have clarified this discrepancy in the method section (see page 4, line 184ff).

In regard to thrombosis only 1 patient was a switch patient and doing per protocol analysis, there was also no difference. The same is valid for bleeding events. You are right that it would be better to account also for the switch group, but due to the low case number we refrained in adding the additional analysis into the manuscript since it does not add valuable additional finding.

Answer 5: The reviewer rightly brings up an important point. Only 5 patients were left until visit 7 and 3 of them fell below 45 seconds, therefore when only looking the patients without LOCF the median aPTT is low with 42 seconds. In fact, only 3 patients remained UFH responders, but since the study treatment was terminated for these patients anyway, they could not have been switched.

Additionally, we have to point out that we analyzed the visits over time with the method of LOCF (last observation carried forward) and since the last aPTT was always below 45 seconds this summed up, but this did not change the aPTT median.

We added figure 4 to indicate the numbers of patients over time and changed the result section accordingly (see page 8, line 296ff). Two of the switch patients were lost to-follow up since they were transferred to another hospital, so that 13 received argatroban. When looking at the numbers in figure 4, one have to keep in mind, that we had some early termination of study treatment. This was the case when the patients had no longer the need for continuous anticoagulation. Per protocol this was not counted as drop out or lost to-follow up, since we could do the safety visits in these patients.

Answer 6: The reviewer interposes an inappropriate wording for argatroban working better than UFH in regards to reaching a predefined anticoagulant goal. We tried to tone that down in all sections accordingly.

Point 7: Rules 294,295 seem to contain a fragment of a previous reviewer's comment?

Answer 7: We apologize for this error. The according lines have been deleted.

We would like to thank this reviewer for the excellent suggestions and that you have really read into our study, which have helped clearly to improve this manuscript.